# A Synthetic Peptide CTL Vaccine Targeting Nucleocapsid Confers Protection from SARS-CoV-2 Challenge in Rhesus Macaques

**DOI:** 10.3390/vaccines9050520

**Published:** 2021-05-18

**Authors:** Paul E. Harris, Trevor Brasel, Christopher Massey, C. V. Herst, Scott Burkholz, Peter Lloyd, Tikoes Blankenberg, Thomas M. Bey, Richard Carback, Thomas Hodge, Serban Ciotlos, Lu Wang, Jason E. Comer, Reid M. Rubsamen

**Affiliations:** 1Department of Medicine, Columbia University, P&S 10-502, 650 West 168th Street, New York, NY 10032, USA; peh1@cumc.columbia.edu; 2Department of Microbiology & Immunology, University of Texas Medical Branch, 301 University Blvd., Galveston, TX 77555, USA; trbrasel@utmb.edu (T.B.); chmassey@utmb.edu (C.M.); 3Flow Pharma Inc., 4829 Galaxy Parkway, Suite K, Warrensville Heights, OH 44128, USA; cvherst@flowpharma.com (C.V.H.); sburkholz@flowpharma.com (S.B.); plloyd@flowpharma.com (P.L.); teakb@aol.com (T.B.); rcarback@flowpharma.com (R.C.); thodge@flowpharma.com (T.H.); sciotlos@flowpharma.com (S.C.); lwang@flowpharma.com (L.W.); 4Dignity Health Mercy Medical Center, Redding, CA 96001, USA; docbey@gmail.com; 5The Department of Anesthesiology and Perioperative Medicine, Case Western Reserve School of Medicine, Cleveland Medical Center, University Hospitals, Cleveland, OH 44106, USA; 6Department of Anesthesia, Critical Care and Pain Medicine, Massachusetts General Hospital, Boston, MA 96001, USA

**Keywords:** SARS-CoV-2, animal model, macaque, vaccine, MHC class I peptide, T cell

## Abstract

Background: Persistent transmission of severe acute respiratory syndrome coronavirus 2 (SARS-CoV-2) has given rise to a COVID-19 pandemic. Several vaccines, conceived in 2020, that evoke protective spike antibody responses are being deployed in mass public health vaccination programs. Recent data suggests, however, that as sequence variation in the spike genome accumulates, some vaccines may lose efficacy. Methods: Using a macaque model of SARS-CoV-2 infection, we tested the efficacy of a peptide-based vaccine targeting MHC class I epitopes on the SARS-CoV-2 nucleocapsid protein. We administered biodegradable microspheres with synthetic peptides and adjuvants to rhesus macaques. Unvaccinated control and vaccinated macaques were challenged with 1 × 10^8^ TCID_50_ units of SARS-CoV-2, followed by assessment of clinical symptoms and viral load, chest radiographs, and sampling of peripheral blood and bronchoalveolar lavage (BAL) fluid for downstream analysis. Results: Vaccinated animals were free of pneumonia-like infiltrates characteristic of SARS-CoV-2 infection and presented with lower viral loads relative to controls. Gene expression in cells collected from BAL samples of vaccinated macaques revealed a unique signature associated with enhanced development of adaptive immune responses relative to control macaques. Conclusions: We demonstrate that a room temperature stable peptide vaccine based on known immunogenic HLA class I bound CTL epitopes from the nucleocapsid protein can provide protection against SARS-CoV-2 infection in nonhuman primates.

## 1. Introduction

As of April 2021, over 1.6% of the world population have had confirmed COVID-19 disease and the new case rate is about half a million per day. Less than 2% of the world’s population are vaccinated against SARS-CoV-2 [1]. Confounding factors in the efforts to reach herd immunity to COVID-19 disease include, but are not limited to, the following: (1) the spread of SARS-CoV-2 mutations affecting the efficacy of current iterations of vaccines and therapeutic biologics [2], (2) the speed of SARS-CoV-2 vaccine deployment, development, and manufacture, and (3) in the context of global public health, issues related to vaccine hesitancy, cold supply-chain requirements, the total manufacturing cost per dose, and ease of administration.

We previously described a novel vaccine platform [3,4], which may address many of the above concerns, and now report on its efficacy in a rhesus macaque model of SARS-CoV-2 infection. Macaque models of COVID-19 disease have been previously reported and have served as critical tools for understanding disease pathology and for the development and testing of vaccines and therapeutics [5,6,7,8,9,10,11,12,13,14,15,16,17,18,19,20,21]. While the clinical course of SARS-CoV-2 infection in macaques is milder relative to that observed in humans [6,8,20,21], the macaque model remains the gold standard for preclinical evaluation of COVID-19 vaccines [10,19,22,23,24,25,26,27]. Our overall approach focuses on promoting protective T-cell immunity using synthetic peptides delivered in biodegradable microspheres together with Toll-like receptor (TLR) 4 and 9 adjuvants and differs from current COVID-19 vaccines against spike proteins. The synthetic peptide sequences applied are based on known immunogenic HLA class I bound epitopes that have previously been characterized in either SARS-CoV-1 [28] or SARS-CoV-2 [29,30,31] infections. We provide evidence that application of this vaccine platform in SARS-CoV-2-challenged macaques provides protection from pneumonia-like pathology observed in virally challenged but unvaccinated control non-human primate (NHP) subjects, reduces viral loads as compared to unvaccinated controls, and induces changes in the gene expression patterns in recovered BAL cells consistent with enhanced antigen presentation capacity and markers of T cells.

## 2. Materials and Methods

### 2.1. Macaque MHC Class I Typing

The MHC class I genes of a cohort of 15 rhesus macaques (Envigo, Alice, TX, USA) were molecularly typed by the University of Wisconsin–Madison National Primate Research Center. Mamu (*Macaca mulatta*) MHC class I alleles were typed amplifying the genomic DNA of each subject using a panel of specific primers for exon 2 of all known MHC-A and MHC-B alleles encoded by each subject’s target DNA. Resulting amplicons were sequenced by the Illumina MiSeq method [32]. The primer panel contained specific primer pairs able to amplify all possible MHC-A and MHC-B alleles encoded by each macaque. Sequencing data analysis provided a high-resolution haplotype for the MHC-A and MHC-B alleles carried by each subject. Following analysis, we selected four of the 15 macaques for vaccination based on peptide MHC binding predictions as described below.

### 2.2. Vaccine Design/Peptide Selection and Manufacture

#### 2.2.1. In Silico Design and Selection of SARS-CoV-2 CTL Epitopes

The overall strategy and rationale for the selection of synthetic peptides used to stimulate potential CTL immune responses in SARS-CoV-2-infected humans has been previously described [3,4]. We selected SARS-CoV-2 nucleoprotein as the target of CTL attack based on the following rationale: (1) survivors of SARS-CoV-1 have shown a memory T-cell response to nucleoprotein at least two years after infection [28], (2) there is >90% amino acid sequence homology between SARS-CoV-1 and SARS-CoV-2 nucleoprotein (the homology for the selected CTL epitopes used in this report is 100%) [33], and (3) in general, there is a lower frequency of mutations resulting in amino acid substitutions (relative to spike protein) that might affect the immunogenicity of the selected CTL epitopes represented by synthetic peptides within the vaccine formulation [34]. The lower mutation frequency may reflect the hypothesis that amino acid substitutions in nucleoprotein may impact viral fitness [35]. We reviewed previous literature and MHC peptide-binding databases [36] and selected five amino sequences representing the SARS-CoV-2 nucleoprotein with predicted strong in vitro affinity for HLA class I molecules [37] and/or documented or predicted immunogenic potential [4,28,29,30,31,33,38,39]. Together this set of peptides yielded potential broad coverage of HLA haplotypes (>90% worldwide) (Appendix A, Appendix A). The predicted binding of this set of peptides was examined within the MHC genotypes of the cohort of 15 rhesus macaques [40] available to us and the best correspondence between selected peptides and the rhesus MHC class I genotype was selected (Appendix A, Appendix A). As the predicted peptide macaque MHC binding coverage for the peptide LLLDRLNQL was incomplete in the available genotypes, we added an additional peptide (ASAFFGMSR) with predicted strong Mamu MHC class I binding to the formulation for a total of six peptides.

#### 2.2.2. Microsphere Preparation and Adjuvant Formulation

The peptide epitopes used in this study were delivered in vivo by intratracheal instillation of a formulation of poly-L-lactide-co-glycolide (PLGA) microspheres containing the corresponding synthetic nine-mer peptides and TLR-9 agonist CpG oligonucleotide adjuvant in a vehicle containing TLR-4 agonist monophosphoryl lipid A (MPLA). The rationale for the choice of the delivery platform and the basic manufacturing scheme used in production has been previously described [3,4,41]. Briefly, room temperature solutions of a synthetic peptide, CpG oligonucleotide, and mannose were mixed with a solution of PLGA in acetone followed by sonication. The formulation was then filtered using a sterile 0.45 micron filter (Pall Corporation, Westborough, MA, USA ), processed through a precision spray-drying device (Buchi Corporation, New Castle, DE, USA), and passed through a drying chamber, with nitrogen gas introduced through a 0.2 micron filter and heated to 65 °C to allow evaporation of the acetone. The dry microsphere stream was analyzed in real time through a laser particle size analyzer (SprayTech, Malvern Instruments, Malvern, PA, USA) before collection (Buchi cyclone dryer) as a dry powder for reconstitution at the time of delivery using a 2% dimethylsufoxide (DMSO) aqueous solution containing MPLA (20 μg/mL). The microspheres and diluent were handled using a BSL-2 non-sterile technique throughout the experiment. Each microsphere contained peptide loaded at approximately 0.1% by weight and CpG 0.01% by weight. Monitoring of the microsphere diameters allowed the production of microspheres with a mean diameter of 10 ± 2 microns. This diameter was selected for formulation to ensure delivery via phagocytosis of no more than 1–4 microspheres per antigen-presenting cell (APC), which have average diameters of 13 microns [3]. cGMP manufacturing protocols were employed using GMP grade synthetic peptides (Peptides International, Louisville, KY, USA), CpG oligodeoxynucleotides (Trilink Biosciences, San Diego, CA, USA), and MPLA (Avanti Polar Lipids, Alabaster, AL, USA). The CpG oligonucleotide and MPLA used in this study were manufactured using the same chemical compositions as equivalent materials used in FDA-approved vaccines. Assessment of thermal stability of the synthetic peptides within the microspheres has been previously reported [3]. Peptide content and structure in microspheres were determined by high-performance liquid chromatography after two months of room temperature storage. We found that over 99% of the peptide was maintained structurally intact.

### 2.3. Animal Studies

#### 2.3.1. Ethics Statement

The animal research protocols used in this study were performed in strict accordance with the recommendations in the Guide for Care and Use of Laboratory Animals, Eighth Edition (National Academy Press, Washington, DC, USA, 2011). The University of Texas Medical Branch (UTMB) facility where these studies were conducted is accredited by the Association for Assessment and Accreditation of Laboratory Animal Care. The protocols were approved by the UTMB Institutional Animal Care and Use Committee (Protocol Numbers 2,004,051 (natural history/control study) and 2,003,033 (vaccination study)) and complied with the Animal Welfare Act, the U.S. Public Health Service Policy, and other federal statutes and regulations related to animals and experiments involving animals. All hands-on manipulations, including immunizations and biosampling, were performed while animals were sedated via ketamine (5 mg/kg)/dexmedetomidine (0.025 mg/kg) intramuscular injection. All efforts were made to minimize suffering.

#### 2.3.2. Macaques

Adult Indian-origin rhesus macaques (*Macaca mulatta*, *n* = 7 (5 male, 2 female), 46–48 months old) or Vietnamese-origin cynomolgus macaques (*Macaca fascicularis*, *n* = 1, female, 84 months old), individually identified via unique tattoo, were obtained from Envigo/Covance (Alice, TX, USA). All animals were considered healthy by a veterinarian before being placed on study. Macaques were individually housed in stainless steel nonhuman primate caging equipped with squeeze backs for the duration of the studies. For continuous core body temperature measurements, a DST micro-T implantable temperature logger (Star–Oddi, Gardabaer, Iceland) was surgically implanted into the peritoneal cavity of each animal prior to study initiation; data recording was set to 10 or 15 min intervals for control and vaccinated macaques, respectively. Certified Primate Diet 5048 was provided to the macaques daily. Drinking water (reverse osmosis-purified) was provided ad libitum through an automatic watering system. To promote and enhance the psychological wellbeing of the animals, food enrichment consisting of fresh fruits and vegetables was provided daily. Environmental enrichment including various manipulatives (Kong toys, mirrors, and puzzles) was also provided.

### 2.4. Immunization, Virus Challenge, Post-Challenge Monitoring and Biosampling

#### 2.4.1. Immunization and ELISPOT Analysis

Immunizations were performed on the selected MHC-typed rhesus macaques (*n* = 4) via ultrasound-guided inguinal lymph node (LN) injection and/or intratracheal instillation (IT). A total of 20 mg of vaccine microsphere preparation in 1 mL was used for each LN injection (two injections/dose/animal) and 100 mg of vaccine microspheres in a 5 mL volume was used for each IT administration. Specifically, on day −51 (51 days prior to virus challenge), two of the macaques (Figure 1) were administered 2 mL of vaccine via LN injection (1 mL per node) and two macaques were administered vaccine via the IT route (5 mls). Subsequent administration of the vaccine, on days −28 and −14, occurred via the IT route only as described previously [42]. On days −44, −21, and −7 (7 days post-vaccination), femoral vein peripheral blood (8 mL) was collected from each animal into a BD Vacutainer^®^ CPT™ Cell Preparation Tube (Becton, Dickinson and Company, Franklin Lakes, New Jersey, USA) with Sodium Heparin (Becton, Dickinson and Company, Franklin Lakes, NJ, USA) and processed to peripheral blood mononuclear cells (PBMCs) per manufacturer instructions. Collected PBMCs were assessed for immunoreactivity via ELISPOT. In brief, ELISPOT assay plates (MabTech Inc., Cincinnati, OH, USA) specific for the detection of primate IFNγ were used according to manufacturer instructions. BAL cell concentrations were adjusted to 1 × 10^5^ cells per mL in a complete growth medium. Diluted BAL cells were dispensed (100 µL/well) into a 96-well plate after which 100 µL of complete growth medium (CGM, negative control), concanavalin A in CGM at 10 µg per well (positive control), and various concentrations of specific (i.e., immunizing) and non-specific peptides (Appendix A) were added. Peptides used for immunization were added to wells at a concentration of 50 µM. All samples were assayed in duplicate. Plates were incubated at 37 °C/5% CO_2_ for 20–22 h, after which plates were thoroughly washed. Conjugated detection antibody was then added and incubated followed by additional washing. Wells were developed using 3,3′,5,5′-tetramethylbenzidine as a substrate. Counts were performed at Cellular Technology Corporation (Shaker Heights, OH, USA) using an Immunospot Analyzer (CTI, Shaker Heights, Ohio, USA) and all well images were quality-controlled on site. All spot-forming cell counts reported are the result of averaging counts from the duplicate 50 µM immunization-specific peptide wells.

#### 2.4.2. Virus Challenge

On day 0, macaques were administered 1–5 × 10^8^ TCID_50_ SARS-CoV-2 (USA_WA1/2020) via combined mucosal atomization (1 mL as delivered using a MAD Nasal™ Intranasal Mucosal Atomization Device (Teleflex, Morrisville, North Carolina, USA) per manufacturer instructions) and intratracheal instillation (4 mL). Intratracheal instillations were performed as described above for delivery of the vaccine. The virus suspension was prepared on the day of challenge from frozen seed stock (kindly provided by Dr. Chien-Te (Kent) Tseng at UTMB) initially generated (one passage) in Vero C1008 (E6) cells (BEI Resources, NR-596, Lot 3956593) from original material provided by the Centers for Disease Control and Prevention in January 2020. Next-generation sequencing confirmed a 100% consensus sequence-level match to the original patient specimen (GenBank accession MN985325.1).

#### 2.4.3. Post-Challenge Monitoring and Chest Radiography

Animals were monitored and scored twice daily for clinical signs of disease, including alterations in activity/appearance (i.e., hunched posture) and food consumption/waste output, and were scored based on general appearance, activity, food consumption, and outward changes in breathing patterns. Prospectively defined criteria that required immediate euthanasia included severe dyspnea and/or agonal breathing and prostate posture/reluctance to move when stimulated. No animals met endpoint criteria during the study. Ventrolateral chest radiography was performed on the days indicated (Figure 1) using a portable GE AMX-4+ computed radiography system (General Electric Company, Boston, Massachussetts, USA) per the manufacturer’s instructions. DICOM data files were independently evaluated by two independent investigators blinded to group assignment with large animal imaging experience via a four-pattern approach (analyses of consolidation, interstitial areas, nodules or masses, and atelectasis).

#### 2.4.4. Biosampling

Blood, nasal cavity samples, and BAL fluid were collected at the indicated times (Figure 1). Femoral vein peripheral blood was collected via Vacutainer^®^ into standard collection tubes containing ethylenediaminetetraacetic acid (EDTA). Hematology was performed on EDTA blood using the Abaxis VETSCAN^®^ HM5 Hematology Analyzer (Abaxis, Inc., Union City, CA, USA). Nasal cavity samples, collected using sterile cotton-tipped medical swabs, were placed into 0.5 mL sterile phosphate-buffered saline (PBS) for viral load analysis. For BAL fluid collection, animals were sedated as previously described and placed in ventral recumbency. The trachea was visualized and cannulated by an appropriately sized rubber feeding tube. Following the placement of the feeding tube, 20 mL of sterile PBS was introduced into the lung and recovered manually through the feeding tube via syringe. This was repeated for a total of 40 mL per animal. The total collected volume from each animal (10–30 mL) was pooled and centrifuged under ambient conditions (10 min at 500× *g*), after which the supernatant was removed. The resulting cell pellet was resuspended in 2 mL of sterile PBS. From this, 1 mL was used for ELISPOT analysis as described for PBMCs. The remaining volume was used for viral load analysis and gene expression profiling.

### 2.5. Viral Load Analysis

#### 2.5.1. Infectious Viral Load (TCID_50_)

Nasal swab and BAL cells suspension samples were serially diluted and incubated with 2 × 10^4^ Vero C1008 (E6) cells (BEI Resources, NR-596, Lot 3956593) in 100 μL of culture medium (MEM/2% FBS) in 96-well flat-bottom plates (*n* = 5 replicate wells per dilution). Each plate contained negative and positive control wells inoculated with culture medium and diluted virus stock, respectively. Cultures were incubated at 37 °C/5% CO_2_ for 96 h after which cytopathic effect was measured via microscopic observation. The TCID_50_/mL value for each sample was calculated as previously described [43].

#### 2.5.2. qRT-PCR

Nasal swab and BAL cell suspension samples (50 μL) were added to TRIzol^®^ LS Reagent (250 μL) and allowed to incubate under ambient conditions for 10 min. Samples were processed to RNA using Zymo Direct-zol™ RNA Mini Prep kits (Zymo Research, Irvine, CA, USA) per manufacturer instructions. RNA samples were analyzed via qRT-PCR targeting the SARS-CoV-2 E gene. The probe (Integrated DNA Technologies, Coralville, IA, USA) was labeled at the 5′-end with fluorophore 9-carboxyfluoroescein (6-FAM) and included an internal quencher (ZEN) and a 3′-end quencher (IowaBlackFQ, IABkFQ). The master mix was prepared by combining forward primer (250 nM, 5′-ACAGGTACGTTAATAGTTAATAGCGT-3′), reverse primer (250 nM, 5′-ATATTGCAGCAGTACGCACACA-3′), and probe (375 nM, 5′-6FAM-ACACTAGCC/ZEN/ATCCTTACTGCGCTTCG-IABkFQ-3′) with 12.5 μL of 2X QuantiFast Probe Mix (QIAGEN, GGermantown, MD, USA), 0.25 μL of 2X QuantiFast RT Mix (QIAGEN), and PCR-grade water (fill to 20 μL). A test sample (5 μL) was added to the master mix, resulting in a final volume of 25 μL per reaction. Real-time analysis was performed using the Bio-Rad CFX96™ Real-Time PCR Detection System. Thermocycling conditions were as follows: step 1, 1 cycle, 50 °C for 10 min; step 2, 1 cycle, 95 °C for 10 min; steps 3–5, 45 cycles, 95 °C for 10 s, 60 °C for 30 s, single read. Negative controls included reaction mixtures without RNA. For quantification purposes, viral RNA extracted from the virus seed stock with a known TCID_50_/mL titer was used. All qRT-PCR results are expressed as TCID_50_/mL equivalents.

### 2.6. Gene Expression Profiling

BAL samples were processed to RNA as described above for qRT-PCR analysis. RNA quantity and quality were assessed using a NanoDrop™ Lite Spectrophotometer (ThermoFisher Scientific, Waltham, MA, USA). Samples, normalized to 20 ng/μL, were analyzed by NanoString Technologies (Seattle, WA, USA) using the nCounter^®^ SPRINT™ Profiler (NanoString Technologies) and gene expression was profiled using the Non-Human Primate Immunology V2 Panel, which contains 754 genes that encompass 17 immune-related signaling pathways with isoform coverage for both *Macaca mulatta* and *Macaca fascicularis*. Probe sets that did not cover both *Macaca* species were eliminated, resulting in a probe set of 730 genes. Raw gene expression data sets received from NanoString Technologies were processed to remove background signals and normalized using the nSolver™ V.3.0 digital analyzer software (NanoString Technologies). Background signal correction was accomplished by subtracting the NanoString negative control genes. Gene expression normalization was performed using the 16 internal reference genes included in the panel.

### 2.7. Study Termination

At scheduled study termination time points (14 and 21 days post-challenge for vaccinated and control macaques, respectively), animals were humanely euthanized via intravenous administration of a pentobarbital-based euthanasia solution under deep anesthesia followed by bilateral thoracotomy.

### 2.8. Statistical Analysis

Descriptive statistics were performed using Microsoft Excel. Hypothesis testing was performed by considering the null hypothesis of the absence of an association between the compared variables. The statistical strength of associations of continuous data was tested using Student’s *t*-testing. Qlucore Omics Explorer 3.5 (Qlucore) and Metascape (Metascape.org) were used to identify the discriminating variables within the NanoString gene expression datasets from BAL sample analysis that were most significantly different between vaccinated and control subjects. The identification of significantly differential variables between the two groups was performed by fitting a linear model for each variable. The set of genes (87 variable genes out of a total of 730 genes) was identified using a *p*-value of 0.05, at least a threefold change, and a q-value cutoff of 0.1. *p*-values were adjusted for multiple testing using the Benjamini–Hochberg method [44]. Gene expression data were scaled to a mean = 0 and a variance = 1 before clustering. Hierarchical clustering of gene expression in BAL was performed using a supervised weighted-average-linkage two-comparison approach. The metric used in scaling dendrogram arms was Pearson’s correlation coefficient.

## 3. Results

### 3.1. Primary Clinical Outcome

#### 3.1.1. Clinical Signs, Body Temperature Alterations, and Hematology

Following SARS-CoV-2 challenge, outward clinical signs measured in control macaques included acute mild lethargy and respiratory distress. All vaccinated animals were normal throughout the post-challenge study period. Core body temperatures, as measured via implanted Star–Oddi DST temperature loggers (Star-Oddi, Reikjavik, Iceland), demonstrated a disruption in the diurnal cycle and mild fever lasting 2–5 days post-challenge in all four control animals (Figure 2, top panels). Conversely, only two of the four vaccinated macaques (RA1693 and RA3797) presented with similar findings, although diurnal cycle disruption was of a shorter duration (1–2 days) and the febrile response was milder (Figure 2, bottom panels). No alterations were measured in the remaining two vaccinated animals. Prior to virus challenge, vaccinated macaques presented with occasional disruptions in the diurnal temperature associated with the vaccination procedure (Appendix A).

Automated hematology analyses were performed on peripheral blood samples (Appendix A). Overall, the number of white blood cells was significantly increased in control subjects on post-challenge days 3 and 5. We observed general lymphopenia in all macaques on day 1 post-challenge. By day 3 post-challenge, lymphocyte counts significantly increased in vaccinated subjects relative to unvaccinated control subjects. Peripheral blood monocyte counts generally peaked on days 1 through 5 in all animals but remained significantly elevated in vaccinated macaques at the end of the study. Neutrophil counts generally rose by day 1 post-challenge in all subjects. There was a transient significant elevation in neutrophils on day 7 post-challenge in vaccinated animals.

#### 3.1.2. Viral Load

Following SARS-CoV-2 challenge, nasal swab and BAL fluid samples were collected throughout the post-challenge period for analysis of infectious viral load and viral RNA via TCID_50_ and qRT-PCR assays, respectively. Infectious virus was measured from nasal swabs of control and vaccinated macaques beginning on day 1 post-challenge (Figure 3, top panel). By Day 7, three of the four unvaccinated animals continued to demonstrate infectious viral shedding, albeit at low levels. In contrast, infectious virus could be measured in only one of the vaccinated macaques at the same time point. By 10 days post-challenge, infectious viral loads were undetectable in nasal swab samples from all animals. Viral RNA in nasal swabs generally reflected infectious viral loads. By day 7, three of the four vaccinated animals demonstrated a 100-fold decrease in nasal swab viral RNA relative to the unvaccinated controls (Figure 3, bottom panel). By 14 days post-challenge, SARS-CoV-2 RNA levels were undetectable in nasal swab samples from all subjects. Infectious viral load data were used to calculate an average viral clearance rate post-challenge for each rhesus macaque. The average viral clearance rate from days 2 through 10 was four- to fivefold higher in two of the four vaccinated macaques (RA1693 and RA3797) relative to the unvaccinated controls (Appendix A).

#### 3.1.3. Macaque Chest Radiography

SARS-CoV-2 challenge in unvaccinated controls resulted in mild-to-moderate lung abnormalities, similar to those previously reported for macaques [6,15,16,20,45,46,47,48]. These were predominantly limited to the caudal lung relative to baseline images, peaked 3–5 days post-challenge, and were qualitatively characteristic of subclinical or mild-to-moderate human COVID-19 (e.g., ground-glass opacities with or without reticulation, paving, silhouetting, and/or linear opacities). The mild-to-moderate interstitial pneumonitis seen on the ventrolateral chest radiographs of unvaccinated subjects are consistent with focal infiltrates representing a complex of interstitial macrophages, neutrophils, and plasmacytoid dendritic cells [49]. Abnormalities in control animals resolved by days 10–21. In contrast, vaccinated macaques lacked the appearance of ground-glass opacities in all regions of the lung throughout the study period (Figure 4 and Appendix A). We did observe, however, modest bilateral increases in reticulation in vaccinated macaques on days 3–5, but these abnormalities also resolved by days 10–21. Bronchoalveolar lavage has been reported to affect computerized tomography X-ray results in healthy rhesus macaques [50]. In our study, however, the pattern of reported changes in vaccinated (i.e., healthy) animals on which BAL was performed was more similar to increases in reticulation versus the patchy consolidations observed in the unvaccinated controls.

### 3.2. Secondary Outcomes

#### Analysis of Gene Expression Patterns in BAL Cells

We identified a set of 87 genes in BAL samples collected 5–7 days post-challenge from control animals with statistically significant differential expression (as measured from changes in accumulation of their specific transcripts) versus BAL samples collected from vaccinated animals during the same time points (Figure 5 and Figure 6). We chose to focus on the day 5 and day 7 samples to capture a possible peak of adaptive immune responses to SARS-CoV-2 challenge, as suggested by previous reports [51,52]. Several of the identified differentially regulated genes were of particular interest in the context of adaptive viral T-cell immunity (Table 1 and Table 2). Several differentially regulated immune response genes lying outside the main window of interest (i.e., day 5 alone, day 7 alone, or day 10 alone) were also identified (Table 1 and Table 2 and Appendix A). For example, on day 5 in unvaccinated macaques, we found upregulation of IFIT3 and IL-1RAP. The expression levels of these transcripts have been previously reported to correlate with viral loads in a SARS-CoV-2 rhesus macaque model of COVID-19 disease [5].

In BAL samples collected on days 5 and 7, we observed statistically significant upregulation of MHC class I genes, MHC class II and associated accessory genes (CD74 invariant chain, HLA-DM), and T-cell markers (CD8 and IL2) in the vaccinated group relative to the unvaccinated control macaques. We also observed statistically significant downregulation of interferon alpha 2 (IFNA2), the negative regulator of T-cell expansion, PD-L1, the decoy receptor for IL1α and IL1β inhibiting signaling, and FoxJ1, a regulator of Th1 cell activation [53], in the vaccinated group relative to the unvaccinated control macaques.. This pattern suggests enhanced antigen presentation and CD 4/8+ T-cell response capacity in BAL cells from vaccinated macaques relative to the controls. In control animals, we observed upregulation of several genes (CCR1, CSF3R, IFNA2, IL-1RN, IL-1RAP, IL-1R2, and SOXS3) previously reported to be activated during SARS-CoV-2 infection in rhesus macaques [5]. These genes appeared to be downregulated, relative to controls, in vaccinated macaques.

The vaccine formulation containing synthetic peptide cytotoxic T cell epitopes, CpG, and MPLA adjuvants was delivered primarily via intratracheal instillation. While we did not observe any adverse clinical signs of respiratory distress in vaccinated subjects, we examined BAL cell gene expression for lymphokines and cytokines associated with the observed mixed Th1/Th2 patterns that have also been observed in asthma [54] (Appendix A). In BAL samples collected on days 5 and 7, we did not observe significant differences in expression of IL4, IL5, IL6, IL9, IL10, IL13, CXCL10, CCL5, CCL7, CCL22, CX3CL1, or CXCL1. We did observe a general trend of higher expression for many of the cytokines in BAL cell samples collected from vaccinated macaques pre-challenge compared to unvaccinated pre-challenge controls. This effect was generally transient, diminishing by day 5 post-challenge. This pattern of cytokine and lymphokine expression did not suggest that BAL cells assumed a phenotype associated with a Th2 response.

A major difference between pre-challenge BAL samples was that vaccinated macaques had received prior intratracheal instillation of vaccine formulation containing CpG and MPLA adjuvants. To assess the possible effects and efficacy of our vaccination procedure and formulation, we compared the expression of 60 genes previously reported to be upregulated not by antigenic stimulation, but by adjuvants alone [59,60,61,62,63]. We found 30 out of 60 genes examined to be significantly (*p* < 0.05) and differentially (>twofold) regulated in BAL cells obtained from vaccinated but unchallenged macaques (day −7) relative to samples from unchallenged control animals (day −1) (Appendix A). This pattern suggests that BAL-associated cells from vaccinated animals were stimulated by the adjuvants in the vaccine formulation. As a further measure of vaccination efficiency, we measured BAL immunoreactivity to the peptides via ELISPOT prior to SARS-CoV-2 challenge (Appendix A). We observed modest immunoreactivity to one of the six CTL peptides (LL9) in three out of the four vaccinated pre-challenge macaques. We did not detect immunoreactivity to the peptide antigens using samples of peripheral blood mononuclear cells (data not shown).

## 4. Discussion

In our study, SARS-CoV-2 infection in unvaccinated control macaques progressed similar to previous reports using this model. The post-viral challenge period was clinically characterized by (1) two waves of infectious viral particle recovery in the nasal tissues, (2) lymphopenia on day 1 post-challenge in all animals, and (3) progressive development of pneumonia-like infiltrations visible on chest X-rays as “ground-glass-like consolidations”, but few changes associated with human SARS-COV-2 infection, such as loss of appetite, respiratory distress, and vomiting and/or diarrhea. We observed that the kinetics of change in viral loads observed in control rhesus macaques were similar to those previously reported [11]. Additionally, the lung tissue abnormalities revealed by chest radiography were similar in the kinetics of progression and depth to previous reports [6,15,16,20,45,46,47,48]. Together, these observations suggest that SARS-CoV-2 infection in the rhesus model is clinically mild, a conclusion confirming some [6,8,11], but not all [64], previous reports. SARS-CoV-2 infection in microsphere/adjuvant-vaccinated macaques progressed in a pattern different from that of the unvaccinated controls. Specifically, disease was characterized by a trend toward diminished recovery of infectious viral particles from nasal tissues, enhanced recovery of peripheral blood lymphocytes counts, and a significant absence of pneumonia-like infiltrates in the lung. Together, these observations suggest that vaccination conferred some degree of protection against SARS-CoV-2-induced disease.

Supporting this clinical conclusion were our studies of the gene expression profiles in serially harvested BAL cells from SARS-CoV-2-challenged macaques and the immunoreactivity of the BAL cells. In samples collected from vaccinated (but pre-viral challenge) macaque BAL cells, we found distinct changes in gene expression associated with the use of TLR 4 and 9 agonists, including upregulation of CSF1 [61], IRF7 [62], and IL10 [60], suggesting effective delivery of the adjuvant portion of the vaccine formulation. When we examined HLA class I restricted immunoreactivity of the pre-challenge but vaccinated macaque BAL cells towards the vaccinating peptides, we found modest-to-low reactivity towards one of the peptides (LLLDRLNQL) in three of four NHP subjects vaccinated.

Following vaccination and viral challenge, we found evidence of upregulation of both Mamu MHC class I and class II genes in macaque BAL cells relative to unvaccinated viral-challenged macaques. Upregulation of HLA class I and class II molecules in peripheral blood mononuclear cells has been reported following the measles/mumps/rubella (MMR) vaccine in MMR-naïve individuals [65]. Increased signatures of M1-type macrophage APC transcripts in the BAL of SARS-CoV-2-infected rhesus macaques has been previously reported [5]; however, our finding of upregulated expression of MHC class II genes, a hallmark of professional APCs, appears to be unique to the BAL of our vaccinated macaques. Likewise, we found increased upregulation of the IL2 genes in vaccinated relative to unvaccinated macaques. A similar finding has been reported in humans, where higher IL 2 levels distinguish mild/asymptomatic forms of COVID-19 disease from the moderate/severe forms [66]. Following viral challenge in our study (specifically, days 5 to 7), we found downregulation of IFNα2 genes in BAL cells recovered from vaccinated subjects relative to unvaccinated controls. The observation perhaps reflects the decreased viral loads found in vaccinated NHP subjects [5].

While antibodies are a critical component of the protective humoral immune response to pathogens, antibodies that promote disease have been described and categorized as ADE [67] or VAERD, such as those described in MERS-CoV or RSV patients [68,69,70]. Since our vaccine only delivered low-molecular-weight nonameric synthetic peptides, unconjugated to any carrier, we did not expect to generate a significant anti-peptide humoral response. Rather, during the development and testing of the microsphere synthetic peptide COVID 19 vaccine, we were mindful of evoking Th2-biased immune responses, particularly those that occur in the absence of Th1 responses or appropriate T regulatory cell responses [71]. The gene expression patterns of BAL cells obtained from the vaccinated macaques in our study suggested that we did not provoke an unbalanced Th2 response by vaccination.

We note several weaknesses in our approach to demonstrate the efficacy of this experimental vaccine: (1) SARS-CoV-2 infection in rhesus macaques resulted in only mild disease, which appeared to resolve by days 10–14 post-infection. Having noted this in previous reports on the SARS-CoV-2/rhesus model, we tried to induce more severe forms of infection using higher doses of infectious particles than previous reports. Based on clinical signs, we found little effect from the increased dose of the virus on rhesus SARS-CoV-2 disease severity. (2) Due to the limited amount of cells recovered from the BAL, we were unable to perform confirmatory qRT-PCR analysis of the unique gene expression patterns found in vaccinated versus control BAL. Nevertheless, we found that many of our observations had been previously reported from studies in similar models. (3) Not all the macaque chest radiographs were performed on the same day in control versus vaccinated subjects, a reflection of the logistic difficulties in working under BSL ¾ conditions. We are confident, however, that we have captured the chest radiographic abnormalities induced by SAR-CoV-2 infection of macaques (i.e., conspicuous consolidations and infiltrations prevalent in the caudal lobe of the right lung) and adequately shown their absence in vaccinated viral-challenged macaques. (4) We only included one cynomolgus macaque in our control group. This was primarily due to macaque availability at the time the study was conducted. However, this appears to have been both a strength [64] and weakness of the experimental design, as we found similar lung pathology in the cynomolgus macaque, as well as similar patterns of BAL gene expression, as for the control rhesus subjects [72]. (5) Finally, we did not study the effects of adjuvant alone on SARS-CoV-2 infection in the rhesus macaques. Previous experience with this microsphere CTL vaccine platform in a murine Ebola virus model has shown that adjuvant alone is not sufficient to confer protection against lethal virus challenge. Protection was conferred only when the corresponding synthetic CTL peptide epitopes were delivered in the microsphere [4].

We believe this report is the first demonstration of efficacy in a preclinical NHP model of SARS-CoV-2 infection of a synthetic peptide-based vaccine based on known and persistently immunogenic HLA class I bound CTL peptide epitopes of SARS nucleoprotein [28]. The SARS-CoV-2 nucleoprotein genomic sequences have shown significantly reduced mutation rates compared to spike protein. As such, it may represent an additional target for vaccination, perhaps in the context of a booster vaccine for use following SARS-CoV-2 spike protein vaccines based on recombinant protein, mRNA, or adenoviral vectors [73,74,75]. The ready ability to change the sequence of the synthetic peptide HLA class I restricted CTL epitopes used in the system is an attractive feature, given the observed rates of mutation in SARS-CoV-2 as it spreads in the human population in the future. A second potentially attractive feature of this vaccine approach is that it can be delivered by aerosolization to the respiratory mucosa, a route previously demonstrated to efficiently generate lung-dwelling tissue-resident memory T cells [76,77].

## 5. Conclusions

We demonstrated that rhesus macaques receiving the microsphere vaccine formulation prior to viral challenge are protected from pneumonia-like lung abnormalities that characterize SARS-CoV-2 infection in unvaccinated control macaques. Analysis of gene expression of cells obtained from bronchiolar lavage showed unique signatures consistent with the hypothesis that vaccination with this platform induces a protective T-cell response in viral-challenged macaques.

## Figures and Tables

**Figure 1 vaccines-09-00520-f001:**
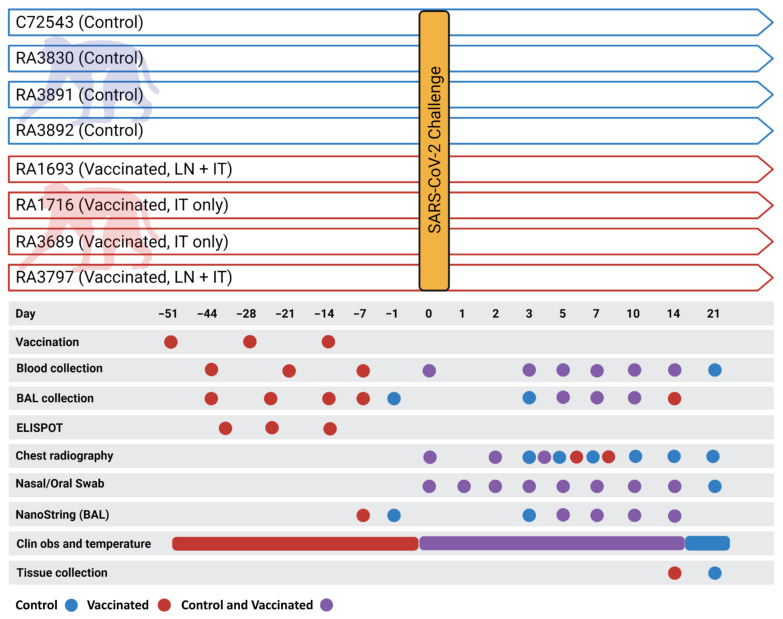
Legend. Schematic of the experimental protocol. Unvaccinated (control) macaques are represented by blue coloring. Vaccinated macaques are represented by red coloring. Overlapping tasks are represented by purple coloring. Graphic created with BioRender.com.

**Figure 2 vaccines-09-00520-f002:**
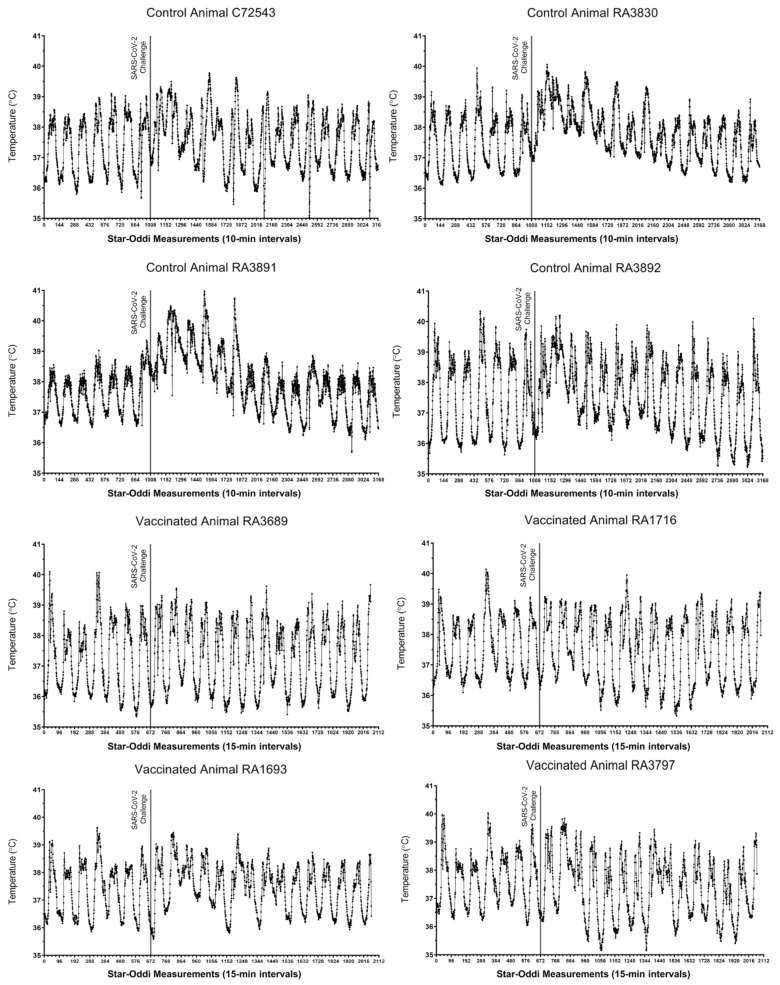
Core body temperature alterations in control and vaccinated macaques following SARS-CoV-2 challenge. For each animal, seven days of pre-challenge baseline temperature measurements are shown. Each tick on the x-axis represents 6 h or 36 individual logger measurements.

**Figure 3 vaccines-09-00520-f003:**
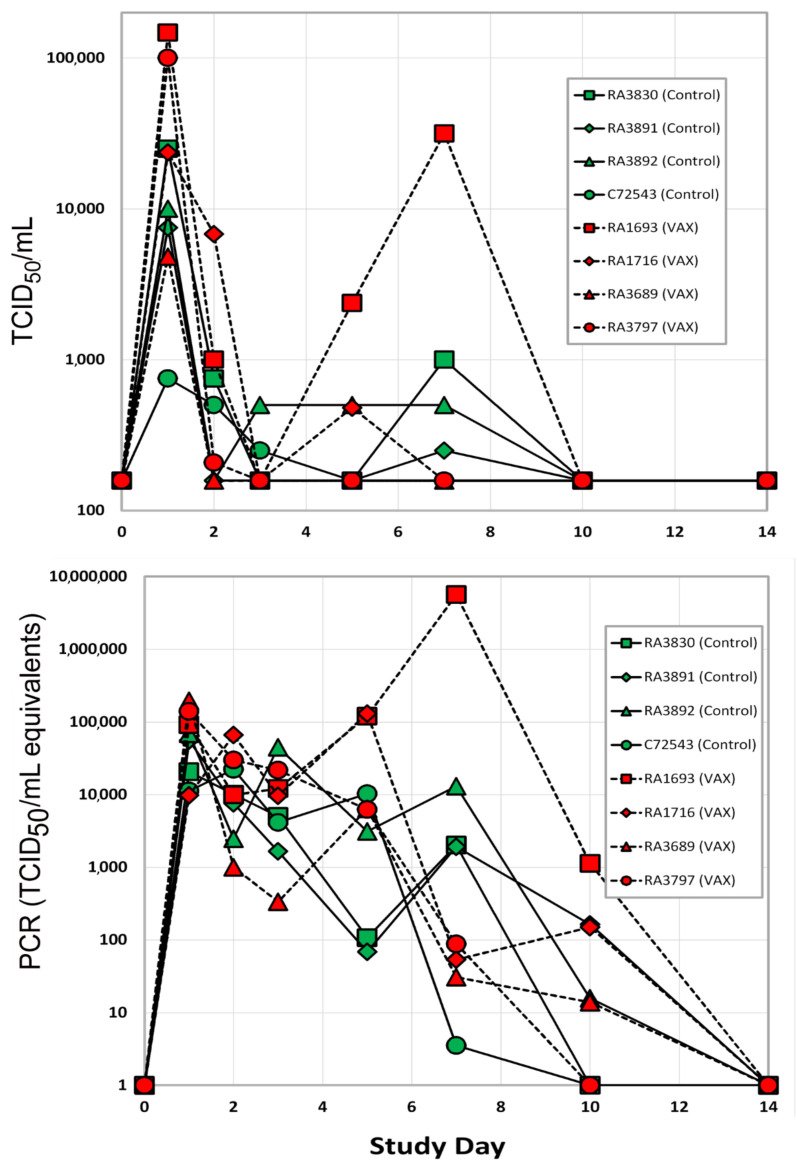
Viral load in nasal swab samples as measured via TCID_50_ assay (**top**) and qRT-PCR (**bottom**). The LLOD of the plaque assay was 150 units. Red symbols are vaccinated rhesus macaques, control unvaccinated rhesus subjects are shown in green symbols.

**Figure 4 vaccines-09-00520-f004:**
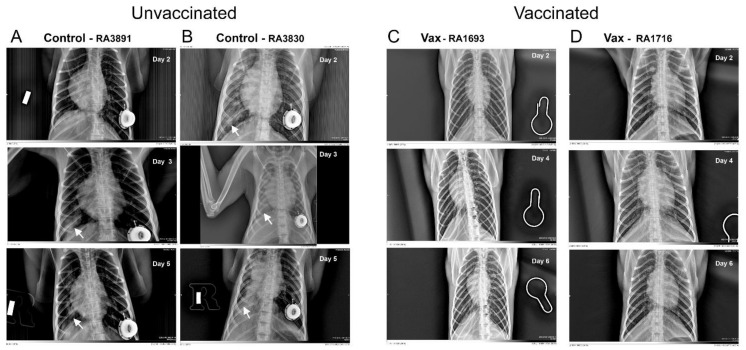
Representative chest radiographs of control and vaccinated macaques following SARS-CoV-2 challenge. As shown, control macaques (left columns (**A**,**B**)) demonstrated a progression of pulmonary infiltrates during the acute period (days 2–5) of disease post-challenge. In contrast, vaccinated macaques (right columns (**C**,**D**)) lacked similar abnormalities. White arrows indicate areas of mild-to-moderate pulmonary infiltrates seen as ground-glass consolidations.

**Figure 5 vaccines-09-00520-f005:**
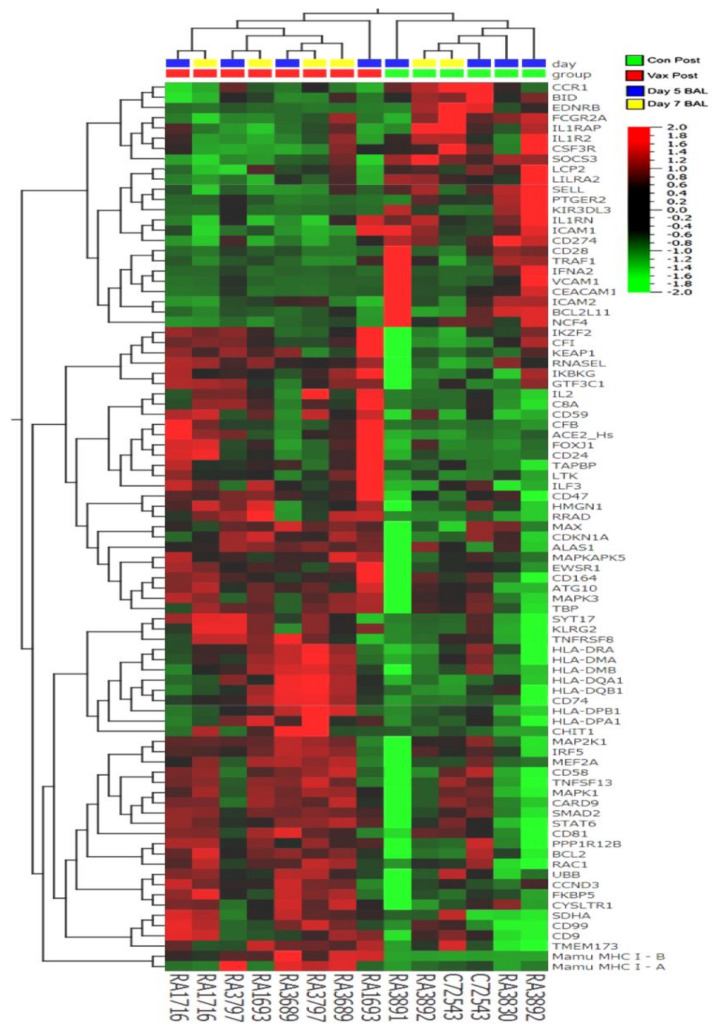
Hierarchical clustering of gene expression in BAL samples collected from control and vaccinated macaques five and seven days post-SARS-CoV-2 challenge. The heatmap shows significantly (*p* < 0.05) upregulated (red) genes (63 genes > threefold) and downregulated (green) genes (24 genes < 1/3-fold) from a total of 730 genes analyzed using the NanoString Non-Human Primate Immunology V2 Panel and identifies a set of genes possibly associated with protection from SARS-CoV-2 challenge.

**Figure 6 vaccines-09-00520-f006:**
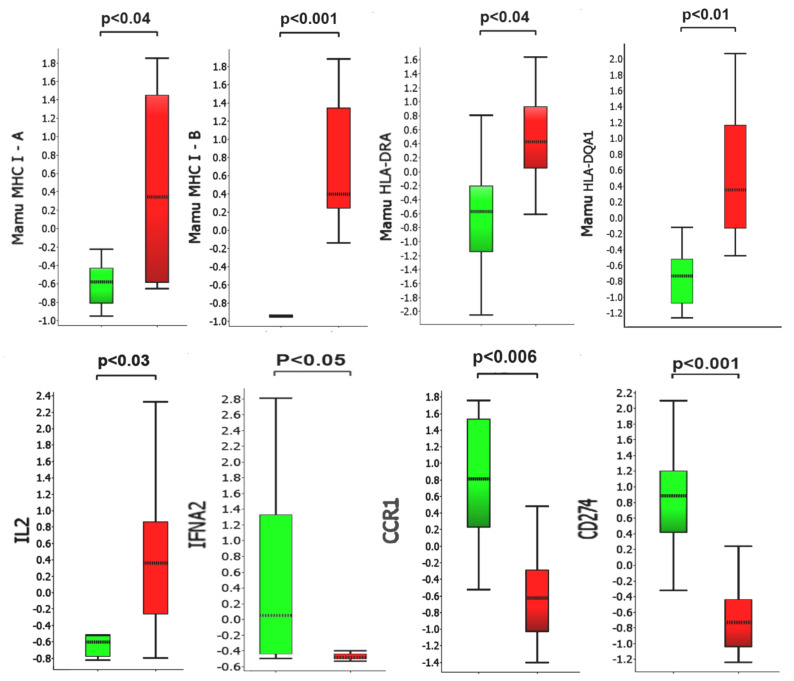
Comparison of selected transcripts up- or downregulated in collected BAL samples 5–7 days post-challenge. Y-axis values represent fold differences in average scaled counts. Green bars and red bars represent control and vaccinated macaques, respectively. *p*-values below 0.05 were considered significant.

**Table 1 vaccines-09-00520-t001:** Differentially regulated genes in BAL cells obtained from SARS-CoV-2-challenged vaccinated versus control macaques.

Upregulated Transcripts on Days 5 and 7 Post-Challenge ^1^					
Gene ID	Function	Distribution	Gene ID	Function	Distribution
Mamu MHC1 A	Antigen presentation to CD8+ T cells^1^	Low cell-type specificity ^1^	CD8	Coreceptor for TCR binding to MHC C1	T cells
Mamu MHC1 B	Antigen presentation to CD8+ T cells	Low cell-type specificity	IL2	Differentiation/maturation of T cells	CD4+ and CD8+ T cells
TAPBP	MHC class I antigen presentation	Low cell-type specificity	CD81	Costimulatory signal with CD3	Low cell-type specificity
HLA-DRA ^2^	Antigen presentation to CD4+ T cells	Professional APC	CD9	Cell adhesion, recognized by CD81	Low cell-type specificity
HLA-DQA1 ^2^	Antigen presentation to CD4+ T cells	Professional APC	CD59	Inhibitor of the complement membrane attack complex	Low cell-type specificity
HLA-DQB1 ^2^	Antigen presentation to CD4+ T cells	Professional APC	CD24	Cell adhesion molecule	Eosinophils and B cells
CD74	MHC class II antigen presentation	Professional APC	CD47	High affinity receptor for thrombospondin-1	Low cell-type specificity
HLA-DMA	MHC class II antigen presentation	Professional APC	CD58	Ligand of the T-lymphocyte CD2 glycoprotein	Low cell-type specificity
HLA-DMB	MHC class II antigen presentation	Professional APC	CD164	Facilitates adhesion of CD34+ cells	Low cell-type specificity
**Upregulated Transcripts on Days 5 or 7 Post-Challenge**			**Upregulated Transcripts on Day 10 Post-Challenge**		
IL17 B	Proinflamatory cytokine	Low cell-type specificity	IL6R	Low affinity receptor for Interleukin 6	Neutrophil
CX3CL1	Chemotactic for T cells and monocytes	Low cell-type specificity	ABL1	Tyrosine-protein kinase, role cell growth and survival	Low cell-type specificity
CD99	Facilitates T-cell adhesion	Low cell-type specificity	TYK2	Tyrosine-protein kinase, initiation of type I IFN signaling	Low cell-type specificity

^1^ Gene annotations supplied by the Human Protein Atlas [55,56]. ^2^ Macaque equivalent of human HLA class II.

**Table 2 vaccines-09-00520-t002:** Differentially regulated genes in BAL cells obtained from SARS-CoV-2-challenged vaccinated versus control macaques.

Downregulated Transcripts on Days 5 and 7 Post-Challenge ^1^					
Gene ID	Function	Distribution	Gene ID	Function	Distribution
IFNA2	Inhibition of viral replication	Macrophages, eosinophils	CD28	Provides costimulatory signals required for T-cell activation Receptor for CD80	T cell
CCR1	C-C chemokine receptor, recruitment of immune effector cells	Macrophages	IL1RAP	Coreceptor with IL1R1 in the IL-1 signaling system	Neutrophil
CD274	Ligand of PD-1 (PDL1), inhibits expansion of antigen-specific CD8+ T cells and CD4+ helper cells	Monocytes, granulocytes	IL1R2	Decoy receptor for IL1α and IL1β (IL1B) inhibiting signaling	Macrophage, neutrophils
**Downregulated Transcripts on Days 5 or 7 Post-Challenge**			**Downregulated Transcripts on Day 10 Post-Challenge**		
CD80	Receptor for CD28 and CTLA-4 on T cells	B cells and monocytes, APCs	HLA-DRA^2^	Antigen presentation to CD4+ T cells	Professional APC
IFNGR2	β chain of the gamma interferon receptor	B cells, APCs and neutrophils	HLA-DMA	Antigen presentation to CD4+ T cells	Professional APC
IL8	C-X-C chemokine for recruitment of neutrophils	Macrophages, epithelial and endothelial cells	LY96	Confers responsiveness to LPS	Macrophages
IL21	Regulates proliferation of mature B and T cells in response to activating stimuli	Activated CD4+ T cells, NKT cells	CTSC	Cathepsin protease	Macrophages
DPP4	Protease upregulated in SARS-CoV-2 [57] Possible viral entry receptor [58]	T-cell CD2	TyroBP	Mediates NK cell activation	Macrophages, monocytes

^1^ Gene annotations supplied by the Human Protein Atlas [55,56]. ^2^ Macaque equivalent of human HLA class II.

## Data Availability

The original DICOM files (>20 MB/file) are available upon request.

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
