# Peer review of "A Synthetic Peptide CTL Vaccine Targeting Nucleocapsid Confers Protection from SARS-CoV-2 Challenge in Rhesus Macaques"

_vaccines, 2021, doi:10.3390/vaccines9050520_

Round 1

Reviewer 1 Report

The work describes the role of a synthetic peptide CTL vaccine against SARS-CoV-2 in rhesus  macaques. The authors support their claim with several experimental approaches. Apart from minor comments, I suggest the publication of the work in the present form.

  1. there are lots of abbreviations in the work ; I suggest the inclusion of a list of abbreviations at the end.
  2. In my opinion I don't think listing all the figures in the supplementary information is  required at the end of the article; anyone interested can visit the supplementary material.
  3. line 172; the sentence should start with 20mg for consistency.

Author Response

Reviewer #1 Comments and Authors reply in italic text

The work describes the role of a synthetic peptide CTL vaccine against SARS-CoV-2 in rhesus macaques. The authors support their claim with several experimental approaches. Apart from minor comments, I suggest the publication of the work in the present form.

  1. There are lots of abbreviations in the work ; I suggest the inclusion of a list of abbreviations at the end.

We thank reviewer for their comments. We have amended the text with the following list of abbreviations used two or more times in the manuscripts.  This list appears on line 636 of the revised MS. 

 Abbreviations used; ADE, antibody mediated disease enhancement; APC, antigen-presenting cell; BAL, bronchoalveolar lavage; CGM, complete growth medium; cGMP, current good manufacturing practice; CpG, cytosine triphosphate deoxynucleotide-guanine triphosphate deoxynucleotide oligodeoxynucleotide; CTL, cytotoxic T lymphocyte; DICOM, digital imaging and communications in medicine; DMSO, dimethyl sulfoxide; EDTA, ethylenediaminetetraacetic acid; ELISPOT, enzyme-linked immune absorbent spot assay; HLA, human leucocyte antigen; IFNϒ, interferon gamma; IT, intratracheal; LLOD, lower limit of detection; LN, lymph node; MHC, major histocompatibility complex; MPLA, monophosphoryl lipid A; NHP, non-human primate; PBMC, peripheral blood mononuclear cells; PLGA, poly-L-lactide-co-glycolide; qRT-PCR, real-time quantitative reverse transcription polymerase chain reaction; SARS-CoV-2, severe acute respiratory syndrome coronavirus 2; TCID50 , fifty-percent tissue culture infective dose; Th1, T helper cell type 1; Th2, T helper cell type 2; TLR, toll-like receptor; VAERD, vaccine associated enhanced respiratory disease.

  1. In my opinion I don't think listing all the figures in the supplementary information is required at the end of the article; anyone interested can visit the supplementary material.

In the MS supplied to authors for revision it appears that all the supplementary information have indeed been moved to the supplementary material for online access

  1. line 172; the sentence should start with 20mg for consistency.

We have made this correction to the MS

Reviewer 2 Report

This study by Harris et al titled “ A synthetic peptide CTL vaccine targeting nucleocapsid confers protection from SARS-CoV-2 challenge in rhesus macaques” targeted MHC class I epitopes of nucleocapsid protein. The current vaccine candidates are targeted against the spike protein and due to sequence variations happening in the spike protein, authors targeted capsid protein. This is a well needed research in current times and I appreciate Harris et al’s work. Coming to the study I have some questions which are pointed below

  1. Why the adjuvant alone was used as a control?
  2. In Figure 2 reading trends of C7253 and RA3830 looked similar to RA3689 and RA1716? In the figure C7253 was mislabeled as C72453.
  3. In figure 3 why is there huge difference between RA1693 and RA3797? I don’t see a significant difference between the controls and vaccinated groups.
  4. In figure 4 were chest radiographs of other macaques taken?

Author Response

Reviewer #2 Comments and Authors reply in italic text

This study by Harris et al titled “A synthetic peptide CTL vaccine targeting nucleocapsid confers protection from SARS-CoV-2 challenge in rhesus macaques” targeted MHC class I epitopes of nucleocapsid protein. The current vaccine candidates are targeted against the spike protein and due to sequence variations happening in the spike protein, authors targeted capsid protein. This is a well needed research in current times and I appreciate Harris et al’s work. Coming to the study I have some questions which are pointed below

Why the adjuvant alone was used as a control?

In the original MS , in the discussion of the weaknesses of our study and on line 533 of the revised MS, we state "5) we did not study the effects of adjuvant alone on SARS-Cov-2 infection in the Rhesus macaques. Previous experience with this microsphere CTL vaccine platform in a murine Ebola virus model has shown that adjuvant alone was not sufficient to confer protection against lethal virus challenge. Protection was conferred only when the corresponding synthetic CTL peptide epitopes were delivered in the microsphere [4]." 

Thus we did not study macaques treated with adjuvant alone and later challenged with virus.  We felt that this aspect of the study was a shortcomming (entirely due to the scarcity of macaques available for study), because formally speaking, our study cannot differentiate between the effects of adjuvants and vaccination by peptide.  However, previous experience in a murine ebola virus challnge model demonstrated that adjuvant alone did not protect the mice from the leathal effects of Ebola challenge. Protection from viral chalenge was confered only when adjuvant and peptide were given. (Herst, C.V.; Burkholz, S.; Sidney, J.; Sette, A.; Harris, P.E.; Massey, S.; Brasel, T.; Cunha-Neto, E.; Rosa, D.S.; Chao, W.C.H., et al. An effective CTL peptide vaccine for Ebola Zaire Based on Survivors' CD8+ targeting of a particu-lar nucleocapsid protein epitope with potential implications for COVID-19 vaccine design. Vaccine 2020, 38, 4464-4475, doi:10.1016/j.vaccine.2020.04.034)

In Figure 2 reading trends of C7253 and RA3830 looked similar to RA3689 and RA1716? In the figure C7253 was mislabeled as C72453.

Complete protection from viral challenge associated alteration in core temperatures was not confered by vaccination.  However, in 2 of 4 vaccinated macaques alteration in core temperatures were almost undetectable.  We have checked the MS and the C72453 designation is correct.  The error is in Figure 1(line 205), now revised, where we incorrectly labelled the first macaque designation. We thank the reviewer for pointing out our mistake!

In figure 3 why is there huge difference between RA1693 and RA3797? I don’t see a significant difference between the controls and vaccinated groups.

The reviewer is correct in pointing out the second wave increase in infectious viral load in nasal swabs from macaque.  We thought that this second wave might be attributed to the highest initial viral load measured on day 1 in RA 1693. Both the TCID50 (Figure 3, top panel) and PCR (Figure 3, bottom panel) suggest that vaccination does not completely ( in 1 out 4 vaccinated macaques) confer a sterilizing immunity on day 7.  Never-the-less, vaccination did confer protection (in 4 out of 4 macaques) from the pneumonia-like radiologic lesions seen in all unvaccinated animals.

In figure 4 were chest radiographs of other macaques taken?

In the MS supplied to authors for revision it appears that all the supplementary information have been moved to the supplementary material for online access. In this section radiographs from all macaques in this study have been provided.  

Reviewer 3 Report

A very interesting article in the context of developing new approaches to developing new vaccine approaches to combat the pandemic. Although the approach is one that allows the vaccine's effectiveness to be unequivocally assessed, I believe that the studies presented here could have been obtained in another type of animal model. The animals could have been kept for a long period and evaluated the time during which the levels of developed immunity are maintained.

The evaluation of the response was quite comprehensive. However, a mucosal assessment of the development of neutralizing antibodies was lacking.

Some specific questions:

How it was possible to produce the powder under sterile conditions?

Is not clear if particles were intact on the administration time and why it was used a solution of 2% of DMSO.

Why were chosen the two vias of administration for immunization?. Why was not used the same via of immunization as the challenge infection?

Why is this referred along the text? “Macaque C75243 (cynomolgus) was not included in this study.”

Author Response

Reviewer #3 Comments and Authors reply in italic text

A very interesting article in the context of developing new approaches to developing new vaccine approaches to combat the pandemic. Although the approach is one that allows the vaccine's effectiveness to be unequivocally assessed, I believe that the studies presented here could have been obtained in another type of animal model. The animals could have been kept for a long period and evaluated the time during which the levels of developed immunity are maintained.

We thank the reviewer for their comments.  We selected the macaque model for these studies because it is the preferred model of the US FDA for preclinical evaluation of vaccines targeted for human use. Indeed, we are in the planning stages of a longer term human phase I study as the reviewer suggests.

The evaluation of the response was quite comprehensive. However, a mucosal assessment of the development of neutralizing antibodies was lacking.

We were of the opinion that since our immunizing material was MHC Class I binding nine-mer peptides corresponding to the SARS-CoV-2 nucleocapsid protein, that the assessment of neutralizing antibodies, those recognizing SARS-CoV-2 spike protein, would be fruitless.   We do acknowledge that if the study had been carried out longer, the provocation of a CTL response to SARS-CoV-2 nucleocapsid protein could result in an enhanced antibody to other SARS-CoV-2 proteins via the mechanism of intermolecular epitope spreading (see reference   Brossart P. The Role of Antigen Spreading in the Efficacy of Immunotherapies. Clin Cancer Res. 2020 Sep 1;26(17):4442-4447. doi: 10.1158/1078-0432.CCR-20-0305. Epub 2020 May 1. PMID: 32357962.).  This study should certainly be part of our future plans and we thank the reviewer.

Some specific questions:

How it was possible to produce the powder under sterile conditions?

We did not prepare the microspheres under sterile conditions for this experiment. However the formulation was filtered aseptically. We have edited the manufacturing section beginning on line 118 as follows:

‘’The formulation was then filtered using a sterile 0.45 micron filter (Pall Corporation) and processed through a precision spray-drying device (Buchi Corporation, New Castle, DE, USA) and passed through a drying chamber with nitrogen gas introduced through a 0.2 micron filter  and heated to  65 degrees C to allow evaporation of the acetone. The dry microsphere stream was analyzed in real-time through a laser particle size analyzer (SprayTech, Malvern Instruments, Malvern, PA) before collection (Buchi cyclone drier) as a dry powder for reconstitution at the time of delivery using a 2% dimethylsulfoxide (DMSO) aqueous solution containing MPLA (20 mg/ml). The microspheres and diluent  were handled using BSL-2 non-sterile technique  throughout the experiment.”

Why were chosen the two vias of administration for immunization?. Why was not used the same via of immunization as the challenge infection?

We selected two different routes for immunization for the first dose as an internal test for selection of the most efficacy.  We only found an ELISPOT response in BAL to the vaccination peptides when we administered the vaccine via the intratracheal route.  The next doses were administered via this intratracheal route.  Additionally, we favored the intratracheal route for its translatability to human vaccination by aerosol inhalation.

Why is this referred along the text? “Macaque C75243 (cynomolgus) was not included in this study.”

This sentence was in error and has been deleted from the MS (lines 258 and 279).